# The Effect of Neurodynamic Techniques on the Dispersion of Intraneural Edema: A Systematic Review with Meta-Analysis

**DOI:** 10.3390/ijerph192114472

**Published:** 2022-11-04

**Authors:** Sergio Nuñez de Arenas-Arroyo, Vicente Martínez-Vizcaíno, Iván Cavero-Redondo, Celia Álvarez-Bueno, Sara Reina-Gutierrez, Ana Torres-Costoso

**Affiliations:** 1Health and Social Research Center, Universidad de Castilla-La Mancha, 16071 Cuenca, Spain; 2Facultad de Ciencias de la Salud, Universidad Autónoma de Chile, Talca 340000, Chile; 3Universidad Artística y Politécnica del Paraguay, Asunción 2024, Paraguay; 4Facultad de Fisioterapia y Enfermería, Universidad de Castilla-La Mancha, 45071 Toledo, Spain

**Keywords:** entrapment neuropathy, nerve compression syndromes, rehabilitation, neurodynamics, neural mobilization, meta-analysis

## Abstract

Background: There is evidence for the positive effects of neurodynamic techniques in some peripheral entrapment neuropathies, but the rationale for these effects has not been validated. We aimed to estimate the direct effect of neurodynamic techniques on the dispersion of artificially induced intraneural edema measured by dye spread in cadavers. Methods: We systematically searched the MEDLINE, WOS, Scopus, and the Cochrane databases from inception to February 2020 for experimental studies addressing the efficacy of neurodynamic techniques on the dispersion of artificially induced intraneural edema. The DerSimonian and Laird method was used to compute pooled estimates of the mean differences (MDs) and its respective 95% confidence intervals (CIs). Subgroup analyses were conducted according to the type of neurodynamic technique. In addition, a 95% prediction interval was calculated to reflect the variation in true treatment effects in different settings, including the effect to be expected in future patients. Results: Pooled results showed a significant increase in fluid dispersion (MD = 2.57 mm; 95%CI: 1.13 to 4.01). Subgroup analysis showed increased dye spread in the tensioning techniques group (MD = 2.22 mm; 95%CI: 0.86 to 3.57). Conclusion: Neurodynamic techniques improved the intraneural edema dispersion and should be considered for the management of peripheral compression neuropathies. Furthermore, tensioning techniques appear to be effective in helping to disperse intraneural edema.

## 1. Introduction

Entrapment neuropathies are caused by the compression and irritation of peripheral nerves as they travel through narrow anatomical spaces [1]. They are characterized by positive sensory signs (gain of function), such as neuropathic pain (increased nerve mechanosensitivity, evoked pain and pain attacks), paresthesias and dysesthesias with an intradermatomal and extradermatomal distribution, as well as negative signs, including loss of function, weakness and atrophy [1,2]. One in ten people will develop an entrapment neuropathy at some point [3], and it is one of the most common causes of neurological consultations in clinical practice [4] due to the high prevalence of entrapment neuropathy.

The development of entrapment neuropathies is usually slow, and neural injury is mostly chronic in nature. Intraneural ischemia is common in mild entrapment neuropathies. Prolonged ischemia can induce a compromised blood supply to the nerve with subsequent edema formation [1] and eventually leads to intraneuronal and extraneuronal fibrotic changes [5]. If this situation persists over time, it may induce demyelination and degeneration of axons, neuroinflammation, changes in axonal transport and in the central nervous system, changes in the central immune-inflammatory mechanism, central sensitization, and changes in cortical representation [1,6].

The first line of treatment for patients with entrapment neuropathies is conservative management, such as physiotherapy or occupational therapy [1,7]. Physiotherapy may include therapeutic exercises, pain education, electrotherapy, biophysical agents, and manual therapy [8]. Neurodynamic techniques are physical therapy methods (manual therapy or therapeutic exercises) that focus on mobilizing peripheral nerves or adopting joint positions that unload the nervous system to facilitate movement between the nervous system and its surrounding structures (interfaces) and reduce the mechanical load on the nervous system to tolerate the compression, friction, and traction forces associated with sport and daily activities [9,10].

When considering the different neurodynamic techniques, a biomechanical distinction can be made between tension and gliding techniques. Although both types of techniques aim to mobilize the nervous system, tension techniques are associated with a significant increase in nerve tension because they use elements of neurodynamic tests that aim to reproduce the patient’s symptoms. In contrast, gliding techniques consist of two or more joint movements in which movements that load the nervous system are simultaneously counteracted by movements that unload the nervous system by mobilizing the nervous system without a substantial increase in tension [10].

There is evidence about the positive effects of neurodynamic techniques on pain and functionality in peripheral entrapment neuropathies related to carpal tunnel syndrome [11], chronic low back pain [12,13], cervicobrachial pain syndrome [12,14], cubital tunnel syndrome [15], osteoarthritis pain of the hand [16], and plantar heel pain [12]. Although it is hypothesized that these techniques could have positive impacts on entrapment neuropathy symptoms by improving intraneural blood flow, axoplasmic flow, and neural connective tissue viscoelasticity and reducing intraneural edema and mechanosensitivity [9], they have not been validated. Thus, the aim of this systematic review and meta-analysis is to estimate the direct effect of neurodynamic techniques on the dispersion of artificially induced intraneural edema in cadavers.

## 2. Materials and Methods

This systematic review was reported according to the Preferred Reporting Items for Systematic Reviews and Meta-Analyses (PRISMA) [17] guidelines and conducted according to the recommendations of the Cochrane Collaboration Handbook [18]. The protocol was registered in the PROSPERO database (registration number: CRD42022299373).

### 2.1. Data Sources and Searches

The MEDLINE (via PubMed), Web of Science and Scopus databases and the Cochrane Database of Systematic Reviews were systematically searched from their inception to 12 February 2022. The search strategy included the following terms: (“nerve therapy” OR “nerve treatment” OR “neural treatment” OR neurodynamic OR neurodynamics OR “manual therapy” OR “nerve stretch” OR “nerve tension” OR “neural tension” OR “nerve mobilization” OR “neural mobilization” OR “nerve glide” OR “nerve gliding” OR “neural glide” OR “neural gliding” OR “nerve gliding exercises” OR neuromobilization OR “neuromobilization maneuver” OR “neurodynamic techniques”) AND (“fluid dispersion” OR “dye spread” OR “intraneural edema” OR “intraneural oedema”). Furthermore, the references of the included articles were manually searched to identify additional eligible studies. The full search strategies for all databases are shown in Appendix A.

### 2.2. Study Selection

Eligible articles were experimental studies (RCTs or non-RCTs and single-arm pre-post studies) that aimed to estimate the efficacy of neurodynamic techniques on artificially induced intraneural edema in cadavers. Studies not written in English or Spanish, review articles, editorials, comments, guidelines, and case reports were excluded.

The literature search, screening, and trial selection were conducted independently by two reviewers (SN-A and AT-C). When there were disagreements, a third researcher made the final decision (IC-R).

### 2.3. Data Extraction and Risk of Bias Assessment

The following data were extracted from the original reports: (i) author information and year of publication; (ii) sample characteristics (country, sample size, age distribution, height, and weight); (iii) intervention characteristics (intervention and control regimen, time of intervention); and (iv) outcomes assessed (fluid dispersion). Data were extracted independently by 2 reviewers (AT-C and SR-G). When there were disagreements, a third researcher made the final decision (IC-R). When necessary, trial authors were contacted up to three times to retrieve missing information.

Two reviewers (SN-A and SR-G) independently assessed the risk of bias using the Cochrane Collaboration’s tool for assessing the risk of bias (RoB2.0) [19]. Any discrepancies were resolved by consensus; a third reviewer (IC-R) resolved any discrepancies if consensus could not be reached. The RoB2.0 tool covers bias in five domains: randomization process, deviations from intended interventions, missing outcome data, measurement of the outcome, and selection of the reported result. Overall, a trial was considered at “low risk of bias” if all domains were judged as “low risk”, “some concerns” if there was at least one domain rated as having “some concerns”, and “high risk of bias” if there was at least one domain judged as “high risk”.

The methodological qualities of the nonrandomized studies and single-arm pre-post studies were assessed using the Risk of Bias in Nonrandomized Studies of Intervention (ROBINS-I) [20]. This tool assesses the risk of bias according to six domains: bias due to confounding, bias in the selection of participants, bias in the classification of interventions, bias due to deviations from the intended interventions, bias due to missing data, and bias in the selection of the reported results. Each domain could be considered low, moderate, or serious, with a critical risk of bias or no information. Overall, bias will be considered a “low risk of bias” if all domains have been classified as a low risk of bias, “moderate risk of bias” if all domains have been classified as a low or moderate risk of bias, “serious risk of bias” if there is at least one domain rated as serious risk, “critical risk of bias” if there is at least one domain rated as critical risk, and “no information” if there is no clear indication that the study is at a serious or critical risk of bias and there is a lack of information about one or more key domains of bias.

### 2.4. Statistical Analysis

The DerSimonian and Laird method [21] random effects model was used to estimate the pooled mean differences (MDs) and 95% confidence interval (95% CI) for the neurodynamic techniques versus the control group (no intervention). MD was calculated for fluid dispersion of intraneural edema as measured by dye spread [22]; positive MD values indicate an improvement in intraneural edema. For single-arm pre-post studies or when studies included two intervention groups, the pooled MD estimate of the control groups from the others that included RCTs was used as the comparison group. Subgroup analyses were conducted according to the type of neurodynamic technique (sliding or tensioning techniques). In addition, a 95% prediction interval was calculated to reflect the variation in true treatment effects in different settings, including the effect to be expected in future patients, which aids clinical decision-making [23].

Heterogeneity was assessed using the I^2^ statistic and the results were categorized as follows: might not be important (0–40%), moderate heterogeneity (30–60%), high heterogeneity (50–90%), very high heterogeneity (75–100%) [18]. Finally, the τ^2^ statistic was calculated to establish the size and clinical relevance of heterogeneity. A τ^2^ estimate of 0.04 can be considered low, 0.14 as moderate, and 0.40 as a substantial degree of the clinical relevance of heterogeneity [24].

A sensitivity analysis was performed using the leave-one-out method [18] to assess the robustness of the summary estimates.

Publication bias was assessed by Egger’s regression asymmetry test [25], and *p* values <0.10 were considered statistically significant. Statistical analyses were performed using R software V.4.1.2 (Boston, MA, USA).

## 3. Results

### 3.1. Systematic Review

Four trials [26,27,28,29] (Figure 1) were included (2 RCTs and 2 pre-post studies), and the reasons for excluding trials are shown in Appendix A. The trials were conducted in two countries: three in the United States and one in Canada and were published between 2011 and 2017. There was a total of 45 cadavers, 19 (42%) of which were female. The cadavers ranged in age from 72 to 81 years, ranged in height from 166 to 173 centimeters, and ranged in weight from 61 to 73 kg (Table 1).

Regarding the intervention regime, two trials compared tensioning neurodynamic techniques with no interventions, one trial compared two different neurodynamic mobilizations (tensioning techniques versus sliding techniques), and one trial was a single-arm pre-post study based on tensioning techniques. Furthermore, in three trials, the intervention duration was 5 min, and one trial reported a 1-min intervention.

Among the studies that applied tensioning techniques, Brown et al. [29] performed a neurodynamic tibial nerve mobilization using passive ankle dorsiflexion and plantar flexion to the end range performed rhythmically with the hip externally rotated and abducted. Boudier et al. [28] performed the median nerve tension technique with the head and neck in maximal contralateral side bending with shoulder depression, arm abduction at 90°, and external rotation with the elbow, wrist, and thumb in full extension. The Gilbert et al. [27] intervention consisted of repetitive neural mobilization of the fourth lumbar nerve root from the resting position to the straight and leg raise test position with 90° hip flexion with the knee/foot in maximal extension and dorsiflexion, respectively. Gilbert et al. [26] performed a simulation of a neurodynamic tension techniques in a cadaveric peripheral nerve section. The trial by Boudier et al. [28] was the only one to perform an intervention with sliding techniques on the median nerve consisting of cycles of full elbow, wrist, and thumb extension, forearm maximal supination, and with ipsilateral head and neck side bending and shoulder girdle depression with the arm at 90° abduction and glenohumeral joint external rotation at 90°. This was followed by thumb, wrist, and elbow flexion past neutral with contralateral head side bending and shoulder girdle depression with the arm at 90° abduction and glenohumeral joint external rotation at 90°.

### 3.2. Risk of Bias Assessment

The overall risk of bias for RCTs, assessed by the ROB 2 tool, showed some concerns for all studies (mainly related to the selection of the reported results) (Appendix A). The overall risk of bias for non-RCTs, assessed by the ROBINS-I tool, showed a moderate risk of bias for all studies (related to the measurement of the outcomes) (Appendix A).

### 3.3. Meta-Analysis

Four studies were included in the meta-analysis comprising five interventions assessing fluid dispersion in intraneural edema artificially induced by the dye spread (Figure 2). Pooled results showed a significant increase in fluid dispersion (MD = 2.57 mm; 95% CI: 1.13 to 4.01) with substantial heterogeneity (I^2^ = 78%, τ^2^ = 1.58, *p* < 0.01). Analyses by type of neurodynamic technique showed increased dye spread in the tensioning techniques group (MD = 2.22 mm; 95% CI: 0.86 to 3.57) with substantial heterogeneity (I^2^ = 62%, τ^2^ = 0.38, *p* = 0.07). Furthermore, sliding techniques appear to have a positive effect on intraneural edema, although the small number of studies did not allow calculation of the pooled MD for these techniques.

### 3.4. Sensitivity Analysis

When one-by-one studies were removed from the analysis to examine the impact of individual studies, the pooled MD estimates for neurodynamic techniques on fluid dispersion did not significantly change (Appendix A).

### 3.5. Publication Bias

There was no evidence of publication bias, as shown by visual inspection of the funnel plot (Appendix A) and Egger´s test (*p* = 0.0058).

## 4. Discussion

This systematic review with meta-analysis provides a synthesis of evidence suggesting that neurodynamic techniques are a suitable intervention to improve fluid dispersion of intraneural edema assessed by dye spread. Furthermore, subgroup analysis by type of neurodynamic technique showed that tensioning techniques effectively increased the dye spread. Sliding techniques also seem to have a positive effect on intraneural edema despite a reduced number of studies.

Clinicians have incorporated neurodynamic techniques to assess and treat compression neuropathies because of their positive effects on signs and symptoms. However, the physiological mechanisms responsible for symptom improvement remain unclear. This systematic review with meta-analysis synthesizes the evidence about the physiological effect of these techniques on fluid dispersion of intraneural edema and justifies the use of these types of techniques to treat entrapment neuropathies.

The compromise of intraneural circulation and axoplasmic flow appears to be the first step in the pathophysiological cascade of nerve injury in entrapment neuropathies. If this situation is maintained, the hypoxia and alterations in microvascular permeability can elicit an inflammatory response in nerve trunks and dorsal root ganglia, leading to endoneurial edema and increased endoneurial fluid pressure. The nociceptors of the nervi-nervorum and sinuvertebral nerves are then sensitized, which contributes to the mechanosensitivity of neuropathic pain. Intraneural edema drainage may be hindered by inadequate lymphatic vessels coursing into the endoneurium and space surrounding the nerve. Our results suggest that, in this situation, neurodynamic techniques could be effective in helping to disperse intraneural edema, thereby preventing progressive intraneural fibrosis and demyelination, axonal degeneration, the formation of abnormal impulse generation sites, and peripheral and central sensitization.

### 4.1. Clinical Implications

The findings of this study suggest that reduction of intraneural edema is a therapeutic effect of neurodynamic mobilization. Although this effect was shown in cadavers, similar effects were reported after one week of neurodynamic intervention in patients with carpal tunnel syndrome [30] consisting of ten repetitions ten times per day of active sliding techniques for the median nerve to maximize the nerve excursion while minimizing an increase in nerve strain. This could explain the effect of the sliding techniques shown in this meta-analysis and could explain the short-term benefits of these techniques in entrapment neuropathies such as neuropathic low back pain [12,13], cervicobrachial pain syndrome [12,14], cubital tunnel syndrome [15], or carpal tunnel syndrome [11]. Given that the intervention consisted of sliding techniques, the results of this systematic review on the possible beneficial effects of this type of technique on intraneural edema are reinforced. Furthermore, the positive effects of tension techniques shown in this meta-analysis coincide with the most recent evidence on these techniques, which suggests that an optimal tension dose applied to the nerve could improve nerve function and pain modulation and could be important in the clinical practice [10].

Although the prediction interval of the results includes zero, wide prediction intervals are common and this interval is mostly on the positive side that favors neurodynamic mobilizations, which should encourage the clinician to use this type of technique in clinical practice [23,31]. However, the effect of some basic parameters of neurodynamic mobilization on intraneural edema, such as the optimal degree of tension, time of intervention, method (passive or active), or long-term effects, have not yet been elucidated. In addition, the different types of neurodynamic techniques have not been sufficiently tested, so it is not possible to make recommendations on the best type of neurodynamic technique. Further research is needed to clarify the efficacy of these techniques based on the aforementioned parameters.

### 4.2. Limitations

First, because of the scarcity of available trials, results of some subgroup analyses should be considered with caution. Second, the risk of bias of the included trials was mainly moderate, and some concerns, such as the selection of the reported results, should not be ignored. Third, the included trials analyzed the short-term effects of neurodynamic techniques on fluid dispersion in cadavers. Therefore, future research needs to analyze the effect of these techniques on humans, and in the mid- and long-term.

## 5. Conclusions

Neurodynamic techniques improved the intraneural edema dispersion and should be considered for the management of peripheral compression neuropathies. Furthermore, tensioning techniques appear to be effective in helping to disperse intraneural edema. Despite the small number of studies, neurodynamic techniques based on sliding could also have a positive effect on intraneural edema.

## Figures and Tables

**Figure 1 ijerph-19-14472-f001:**
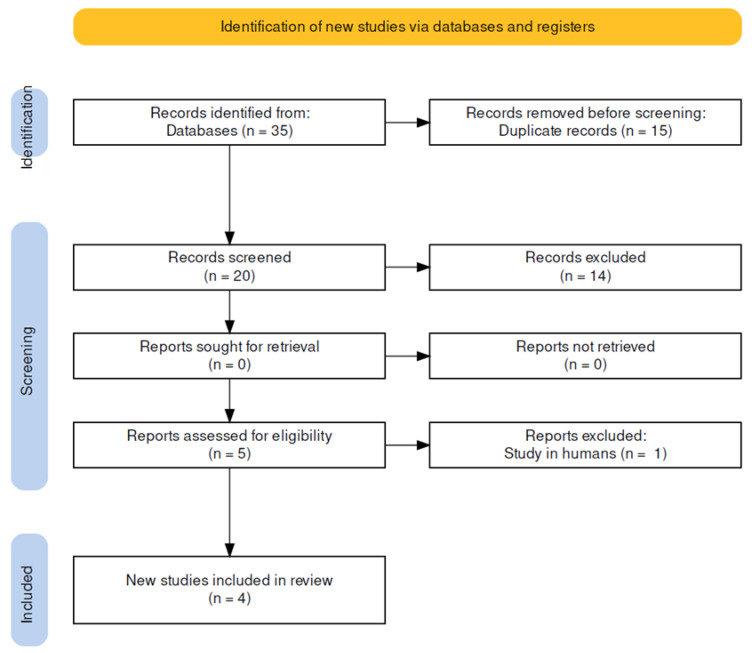
Flow chart.

**Figure 2 ijerph-19-14472-f002:**
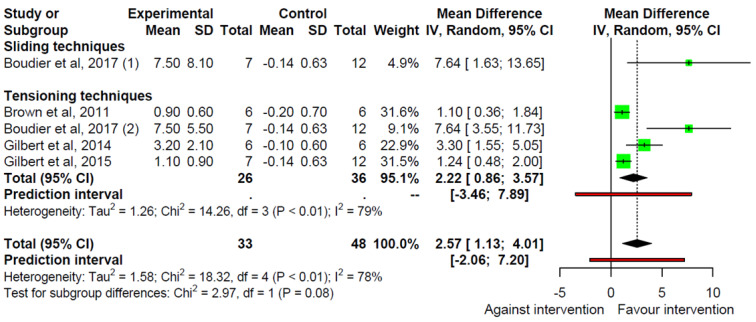
Forest plot showing the mean differences for the fluid dispersion [26,27,28,29].

**Table 1 ijerph-19-14472-t001:** Characteristics of the studies included in the meta-analysis.

Study Characteristics		Sample Characteristics				Intervention Characteristics		
First Author, Year of Publication	Country	Sample Size (%Female)	Age (Years)	Height (cm)	Weight (kg)	Intervention	Time of Intervention (min)	Outcomes
Boudier et al., 2017 [28]	Canada	IG1: 7 (28.6) IG2: 7 (28.6)	81 ± 12	167.0 ± 11	61.0 ± 6.3	IG1: Nerve tensioning techniques. IG2: Nerve sliding techniques.	5	Fluid dispersion
Brown et al., 2011 [29]	USA	IG: 6 (50.0) CG: 6 (50.0)	72 ± 10.36	166.0 ± 12.7	66.0 ± 15.5	IG: Nerve tensioning techniques. CG: No intervention.	1	Fluid dispersion
Gilbert et al., 2015 [27]	USA	IG: 7 (57.1)	74 ± 10.3	173.8 ± 8.9	73.1 ± 12.0	IG: Nerve tensioning techniques.	5	Fluid dispersion
Gilbert et al., 2014 [26]	USA	IG: 6 (50.0) CG: 6 (50.0)	72 ± 10.36	166.0 ± 12.7	66.0 ± 15.5	IG: Nerve tensioning techniques. CG: No intervention.	5	Fluid dispersion

Values are presented as mean ± SD unless otherwise indicated. Abbreviations: cm: centimeters; CG: Control group; Kg: Kilograms; IG: Intervention group; min: minutes.

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
