# Peer review of "The Effect of Neurodynamic Techniques on the Dispersion of Intraneural Edema: A Systematic Review with Meta-Analysis"

_ijerph, 2022, doi:10.3390/ijerph192114472_

Round 1

Reviewer 2 Report

ijerph-1940926

The effect of neurodynamic techniques on the dispersion of intraneural oedema: A systematic review with meta-analysis

This article is straightforward and easy to read. It focuses on a common neurological problem with interest to orthopedic and neurological surgeons, too.

Unfortunately, the number of meta-analyzed studies is rather low.

Minor points:

Abstract, line 17: ... we ... searched ... past tense

Major points:

1.    A more detailed comparison of the excluded study [Schmid et al.] in "living" humans compared to your group would have been a most interesting point to discuss. It strenghthens your results concerning the clinical implications.

Figures

Figure 1: The legend doubles Figure 1. How can you identify 4 records and exclude 15 ?

Figure 2 is not clearly lisible in the present layout.

Table 1: is not cited in the text. The legend is confusing / inadequate. Is there a standard deviation to age / height / weight ?
